# Network conditioning for synergistic learning on partial annotations

**Benjamin Billot**[1]                                                    BBILLOT@MIT.EDU
**Neel Dey**[1]                                                              DEY@MIT.EDU
**Esra Abaci Turk**[2]                          ESRA.ABACITURK@CHILDRENS.HARVARD.EDU
**P. Ellen Grant**[2]                            ELLEN.GRANT@CHILDRENS.HARVARD.EDU
**Polina Golland**[1]                                              POLINA@CSAIL.MIT.EDU

[1] *Massachusetts Institute of Technology, USA*

[2] *Boston Children's Hospital and Harvard Medical School, USA*

**Editors:** Accepted for publication at MIDL 2024

## Abstract

The robustness and accuracy of multi-organ segmentation networks is limited by the scarcity of labels. A common strategy to alleviate the annotation burden is to use partially labelled datasets, where each image can be annotated for a subset of all organs of interest. Unfortunately, this approach causes inconsistencies in the background class since it can now include target organs. Moreover, we consider the even more relaxed setting of region-based segmentation, where voxels can be labelled for super-regions, thus causing further inconsistencies across annotations. Here we propose CoNeMOS (Conditional Network for Multi-Organ Segmentation), a framework that leverages a label-conditioned network for synergistic learning on partially labelled region-based segmentations. Conditioning is achieved by combining convolutions with expressive Feature-wise Linear Modulation (FiLM) layers, whose parameters are controlled by an auxiliary network. In contrast to other conditioning methods, FiLM layers are stable to train and add negligible computation overhead, which enables us to condition the entire network. As a result, the network can *learn* where it needs to extract shared or label-specific features, instead of imposing it with the architecture (e.g., with different segmentation heads). By encouraging flexible synergies across labels, our method obtains state-of-the-art results for the segmentation of challenging low-resolution fetal MRI data. Our code is available at https://github.com/BBillot/CoNeMOS.

**Keywords:** Conditional layers, Partially supervised learning, Region-based segmentation

## 1. Introduction

Multi-organ segmentation is paramount in medical imaging as it is used in clinical practice for surgery planning, and enables volumetric and morphological research studies. However, state-of-the-art learning-based methods, such as convolutional neural networks (CNN), necessitate hundreds of labelled training examples to produce robust and accurate segmentations during inference. Such quantities of data are often not available since manual annotations are costly. Hence, there is a high interest in developing training strategies that can learn from limited supervision in order to reduce the overall annotation effort.

One solution is to combine fully annotated (generally small) datasets with large amounts of unlabelled images. This strategy is implemented in semi-supervised learning, where unsupervised data is used to learn structured feature spaces for improved consistency (Chen

et al., 2020). Another line of research, known as weakly-supervised learning, seeks to leverage cheaper annotations to still benefit from supervision while reducing the labelling cost. This has been achieved by using image-level tags (Papandreou et al., 2015), point annotations (Bearman et al., 2016), or scribbles (Lin et al., 2016). Here, we consider a third scenario, known as *partially supervised learning* (PSL), where images can be segmented for different subsets of all organs of interest. The key idea is to train unified multi-organ segmentation models that learn robust data representations by leveraging the complementary information from the supervision of individual labels. Despite requiring less labelling labour, PSL is challenging algorithmically because it creates inconsistencies in the background class, which can contain target organs. Moreover, we adopt the general case of *region-based segmentation*, where the same voxel can have several labels corresponding to coarser/finer levels of annotations. Region-based segmentation fits well with partial supervision since it enables us to use coarse super-classes to alleviate the annotation burden, but introduces further inconsistencies across training segmentations.

Mainstream approaches avoid such inconsistencies by training a separate network for each region (Zhu et al., 2019; Zhang et al., 2019). However, this technique not only scales poorly with increasing numbers of regions, but also fails to benefit from inter-label information. Hence, PSL-aware losses have been proposed to train unified networks (Roulet et al., 2019; Fidon et al., 2021), but these are difficult to extend to region-based segmentations. Meanwhile, architectural designs have also been proposed to handle PSL. These explicitly model cross-label synergies by sharing parts of the network while keeping others region-specific (Chen et al., 2020; Xu et al., 2023). Finally, Zhang et al. (2021) have proposed a network that segments one region at a time using an adaptive segmentation head. Building on these results, we present a generalised network conditioning approach for PSL that can handle partial labels with possible overlaps between regions.

**Contributions.** We present CoNeMOS (Conditional Network for Multi-Organ Segmentation), a framework that leverages a conditional network for synergistic learning on partially labelled region-based segmentations. We capture cross-label synergies with a shared common segmentation network whose intermediate representations dynamically adapt to the assigned label via an auxiliary network that encodes rich label-specific information. As such, labels are segmented on-demand, which particularly suits the region-based segmentation scenario, since predictions can be retrieved at a desired level of details without having to compute the rest of the labels. In contrast to Zhang et al. (2021), we use a conditioning strategy that applies to *all* network layers. This is achieved by using Feature-wise Linear Modulation (FiLM) layers (Perez et al., 2018), which add negligible computation overhead and are stable to train. As a result, conditioning the entire network enables us to *learn* where inter-label features should be shared, rather than rigidly enforcing it in the architecture.

We demonstrate CoNeMOS on challenging 3D fetal MRI data with low tissue contrast and resolution. In addition to the complex and evolving fetal anatomy, learning robust cross-label representations is challenging in this application due to the large morphological differences across organs of interest (e.g., thin and curvy placenta, deformable uterus, small round brain). Overall, CoNeMOS is comprehensively evaluated in three experiments, where it outperforms the current state-of-the-art PSL conditioning method (Zhang et al., 2021), as well as multi-task learning and pseudo-labelling approaches.

## 2. Related work

**Partially supervised learning (PSL)** handles training data with partial labels. One approach is to train a unified network by developing PSL-aware losses that account for partial labels. In the case of mutually exclusive labels, the loss function can dynamically merge the predictions of unlabelled organs with the background (Roulet et al., 2019; Fidon et al., 2021; Shi et al., 2021; Dorent et al., 2021; Atzeni et al., 2022). However, these losses do not extend to region-based segmentations for which current techniques simply compute the loss on the available labels (Ulrich et al., 2023). Alternatively, partial supervision can be addressed by volumetric regularisation (Zhou et al., 2019). While this method is compatible with region-based segmentations (Ulrich et al., 2023), it implicitly assumes that the volumes of the organs change little in the target cohort, which might not suit the rapidly evolving fetal anatomy. Meanwhile, semi-supervised learning has also been proposed as a way to mitigate partial annotations including pseudo-labelling strategies (Bai et al., 2017; Zhou et al., 2019), weighted average models (Huang et al., 2020), or mean teacher models (Filbrandt et al., 2021). Although not the focus of this work, we note that the proposed conditional architecture can be readily combined with semi-supervised learning approaches.

PSL can also be addressed by explicitly modelling cross-label synergies in the architecture. Multiple methods have proposed to formulate PSL as a multi-task learning problem, with label-specific segmentation heads (Chen et al., 2019; Ulrich et al., 2023), possibly combined with multi-scale attention blocks (Fang and Yan, 2020; Hongdong et al., 2022). In contrast, a recent method posed PSL as a harmonisation task with label-specific encoders (Xu et al., 2023). Closer to our work, partial supervision has also been tackled with conditional networks, initially using hash tables (Dmitriev and Kaufman, 2019), and more recently with a dynamic segmentation head conditioned on the target label and on extracted image features (Zhang et al., 2021). While this last method yields state-of-the-art results for PSL, conditioning the segmentation on the image revealed to be unstable on our low SNR fetal data. Overall, instead of imposing shared data representations in specific parts of the network (e.g., encoder, decoder, last layer), we propose to flexibly learn where synergies are best captured throughout the entire network.

**Conditioning** enables to dynamically adapt networks based on metadata. HyperNetworks (Ha et al., 2017) are the most prominent conditional architecture, where the weights of a main CNN are predicted by an auxiliary multi-layer perceptron (MLP). This design has been applied to a variety of tasks including segmentation of natural (Nirkin et al., 2021) and medical images (Ma et al., 2022). However, given the vast numbers of weights to predict, HyperNetworks have been consistently reported to be slow and unstable to train (Ortiz et al., 2023). For this reason, lighter conditioning methods have been proposed, such as adaptive instance normalisation (Huang and Belongie, 2017), feature-wise scaling (Chen et al., 2020; Tian et al., 2020) and linear modulation (FiLM) (Perez et al., 2018). Owing to their negligible computation overhead and high expressiveness, FiLM layers have shown promising results on medical images for modality transfer (Chartsias et al., 2020), atlas building (Dey et al., 2021), and tumour segmentation based on patient metadata (Lemay et al., 2021). Here, we apply FiLM layers to capture cross-label synergies by modulating shared convolutional layers with label-specific information.

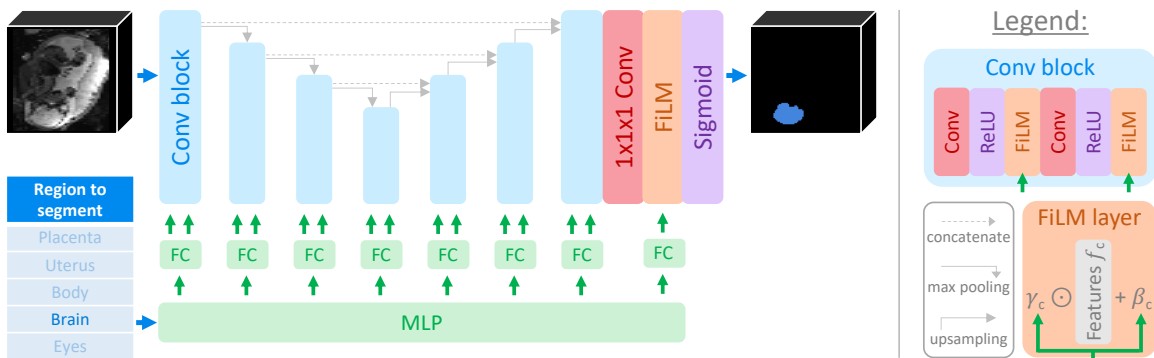

Figure 1: Overview of CoNeMOS. Regions are segmented on demand using a network with shared convolutions. Region-specific information is dynamically injected using FiLM layers, whose parameters are produced by a region-conditioned MLP.

## 3. Methods

### 3.1. Problem definition

Let us define a dataset $\mathcal{D} = \{\mathcal{X}, \mathcal{Y}\}$ of $N$ 3D images $\mathcal{X} = \{X_n\}_{n=1}^N$ with segmentation maps $\mathcal{Y} = \{Y_n\}_{n=1}^N$ annotated for $K$ structures of interest. Further, let us assume that all images are padded to the same height $H$, width $W$, and depth $D$, such that $X_n \in \mathbb{R}^{H \times W \times D}$, for all $n \in \{1, ..., N\}$. Here, we consider the case of *partially supervised* data, where each label map $Y_n$ may include annotations for a subset of all $K$ labels. In other words, for each $Y_n$, we define two sets $\mathfrak{L}_n$ and $\mathfrak{U}_n$ of labelled and unlabelled structures, such that $\mathfrak{L}_n \cup \mathfrak{U}_n = \{1, ..., K\}$. Full supervision is attained when $\mathfrak{U}_n = \varnothing$, $n = 1, ..., N$.

In the practical scenario of *region-based* segmentation, the voxels of $Y_n$ can be assigned to several labels (i.e., *regions*). We emphasise that we consider the most general case, where we do not assume any hierarchy between regions (i.e., any two regions can be disjoint, overlapping, or nested). In this scenario, we assume $Y_n$ is a multi-channel segmentation map, i.e., $Y_n = \{Y_{nk}\}_{k=1}^K$, where $Y_{nk} \in \mathbb{R}^{H \times W \times D}$ holds binary annotations if $k \in \mathfrak{L}_n$, or is empty if $k \in \mathfrak{U}_n$. Finally, let us re-arrange the available binary region annotations into $K$ subsets, i.e., $\mathcal{D} = \{\mathcal{D}_k\}_{k=1}^K$ where $\mathcal{D}_k = \{\{X_n, Y_{nk}\}, n \in \{1, ..., N\}, k \in \mathfrak{L}_n\}$.

### 3.2. Training segmentation networks on partial annotations

Our goal is to learn multi-organ segmentation given a partially annotated dataset. A simple baseline approach is to train separate networks $F_k$ (parametrised by $\theta_k$) for each region $k$:

$$\hat{\theta}_k = \underset{\theta_k}{\operatorname{argmin}} \ \mathbb{E}_{\{X, Y_k\} \sim \mathcal{D}_k} \ \mathcal{L}\left[F_k(X|\theta_k); Y_k\right], \quad k \in \{1, ...K\}, \tag{1}$$

where $\mathcal{L}$ is a loss that compares predictions $F_k(X|\theta_k)$ with ground truths $Y_k$. This strategy scales poorly with the number of labels and does not leverage mutually beneficial cross-label information. An alternative solution is to train a *unified* network $F$ (possibly with label-specific parts) with parameters $\theta$, which can be achieved by using a PSL-aware loss $\mathcal{L}_{\text{PSL}}$:

$$\hat{\theta} = \underset{\theta}{\operatorname{argmin}} \ \mathbb{E}_{\{X, Y\} \sim \mathcal{D}} \ \mathcal{L}_{\text{PSL}}\left[F(X|\theta); Y\right]. \tag{2}$$

A straightforward method for constructing $\mathcal{L}_{\mathrm{PSL}}$ is to consider only the labelled regions $\mathfrak{L}_n$. More elaborate strategies also include $\mathfrak{U}_n$ when computing the loss (Roulet et al., 2019; Fidon et al., 2021; Filbrandt et al., 2021), but these only apply to mutually exclusive labels. Here, we propose a third approach based on conditioning, where regions are segmented *on demand* using a common network that is dynamically modulated for the requested label:

$$\hat{\theta} = \underset{\theta}{\mathrm{argmin}} \ \mathbb{E}_{k\sim\{1,\dots K\}} \ \mathbb{E}_{\{X,Y_k\}\sim\mathcal{D}_k} \ \mathcal{L}\left[\, F(X, k|\theta); \ Y_k \,\right]. \tag{3}$$

This strategy is compatible with region-based labels, since we only compute a marginal loss for one label at a time, and also encourages cross-label synergies while benefiting from label-specific information.

### 3.3. Proposed architecture

The proposed framework, coined CoNeMOS, relies on a conditional architecture that models both cross-label and label-specific information (Figure 1). A backbone network $F$, here implemented as a UNet (Ronneberger et al., 2015), is in charge of the segmentation task. Since all convolutional layers are shared across labels, $F$ enables us to build cross-region synergies. Label-specific information is injected by using an auxiliary network $G$ (a shallow MLP) that dynamically modulates the intermediate features of $F$ based on the region to segment. This is achieved by passing the target label as a one-hot vector $z$ to $G$, which predicts region-specific parameters $G(z)$ to condition the activations of $F$. Specifically, we implement the conditioning mechanism as FiLM layers (Perez et al., 2018). If $f_{lc}$ is the $c$-th output channel of the $l$-th convolutional layer of $F$, then $FiLM(f_{lc}) = (1 + \gamma_{lc}(z)) * f_{lc} + \beta_{lc}(z)$. Here, $\gamma_{lc}(z)$ and $\beta_{lc}(z)$ are scaling and shifting parameters, produced by the MLP $G(z)$, that enable an expressive modulation of the intermediate representations of $F$. Overall, we use FiLM layers throughout $F$ to learn where features should be shared (i.e., $\gamma_{lc}$ and $\beta_{lc}$ independent of $z$), or kept label-specific (i.e., $\gamma_{lc}$ and $\beta_{lc}$ vary significantly with $z$).

### 3.4. Training and inference

CoNeMOS is trained end-to-end as described in Equation (3) with a soft Dice loss (Milletari et al., 2016). Importantly, $k$ is sampled inversely proportionally to the size of $\mathcal{D}_k$, to alleviate class imbalance issues inherent to PSL since regions that are hard to segment might be less frequent in $\mathcal{D}$. At test-time, regions are segmented on demand by modifying $z$.

### 3.5. Implementation details

**Augmentation.** We perform extensive data augmentation to improve the model's robustness. This first includes linear and elastic spatial transforms to increase the morphological variability seen at training (Hoffmann et al., 2022). Intensities are then augmented with bias field corruption, blurring, $\gamma$-exponentiation, and noise injection (Billot et al., 2023).

**Architecture.** $F$ is a light 3D UNet of 4 levels, each using 2 convolutions with 16 kernels of size 3×3×3, ReLU activation, and FiLM modulation. Binary soft segmentations are obtained by $F$ after the final sigmoid activation. $G$ is a shallow MLP of 4 dense layers (64 units each) separated with Leaky-ReLUs (slope of 0.2). FiLM parameters $\gamma_{lc}$ and $\beta_{lc}$ are obtained separately for each convolutional layer of $F$ from fully-connected layers of $2\times16=32$ units.

Table 1: Data splits in terms of fetal scans and corresponding partial annotations.

|                    | Number of scans | Uterus | Body | Brain | Eyes | Placenta |
|--------------------|-----------------|--------|------|-------|------|----------|
| Training (60%)     | 174             | 12     | 12   | 76    | 69   | 106      |
| Validation (15%)   | 42              | 3      | 3    | 18    | 16   | 26       |
| Testing (25%)      | 73              | 5      | 5    | 32    | 29   | 44       |
| Total              | 289             | 20     | 20   | 126   | 114  | 176      |

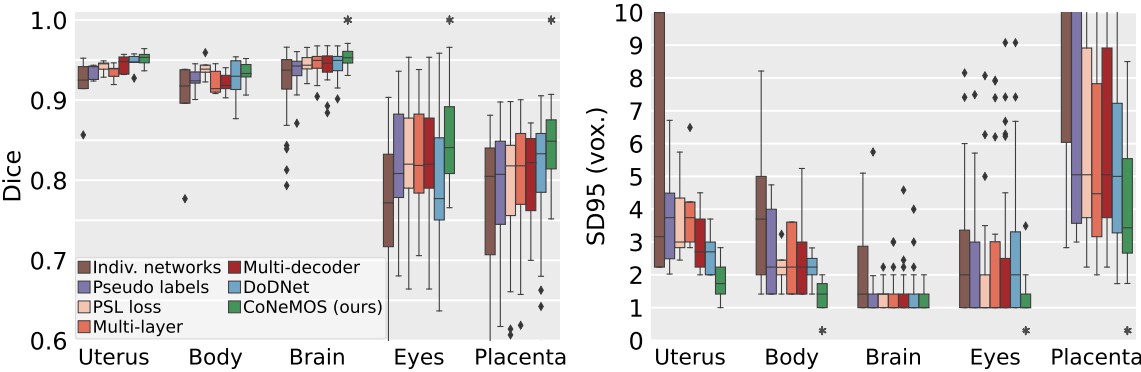

Figure 2: Dice scores and SD95 (95$^{\text{th}}$ percentile of surface distance). Stars $*$ denote statistical significance at a 5% level (Bonferroni-corrected Wilcoxon signed-rank test).

## 4. Experiments and results

### 4.1. Experimental Set-up

**Dataset:** We demonstrate CoNeMOS on an in-house fetal dataset of 289 3D whole-uterus EPI MRI scans from 91 pregnant mothers. Scans are acquired on a 3T Skyra Siemens scanner using multi-slice gradient echo EPI sequences at 3mm isotropic resolution (TR = [5-8]ms, TE = [32-38]ms, $\alpha = 90°$). All scans are padded to a $128^3$ grid, and intensities are rescaled to [0,1]. These scans are partially annotated for 5 regions: uterus (N=20), fetal body (N=20), fetal brain (N=126), fetal eyes (N=114), and placenta (N=176). As detailed in Table 1, we use splits of 60%, 15%, and 25% for training, validation, and testing, respectively.

**Baselines:** We first compare CoNeMOS against **individual networks** trained separately on each region. Then, we assess a **pseudo-labelling** strategy (Filbrandt et al., 2021) to train a unified network. Further, we use a PSL-aware loss, which computes the loss only on available labels, to train a unified network (**PSL loss**, Fidon et al. (2021), Appendix A) as well as two multi-task learning frameworks with label-specific decoders (**Multi-decoder**, Chen et al. (2019)) or last layers (**Multi-layer**, Ulrich et al. (2023)). Finally, we evaluate **DoDNet** (Zhang et al., 2021), the state-of-the-art method in PSL, which relies on a dynamic segmentation head conditioned on the target region and extracted image features.

For fairness, all methods use the same backbone architecture and augmentation scheme, optimised on the validation set. Networks (Tensorflow) are trained for 50,000 steps with the Adam optimiser ($10^{-5}$ learning rate). We train all methods twice, and chose final models based on validation scores. Training takes 1 day on a Nvidia TitanXP GPU.

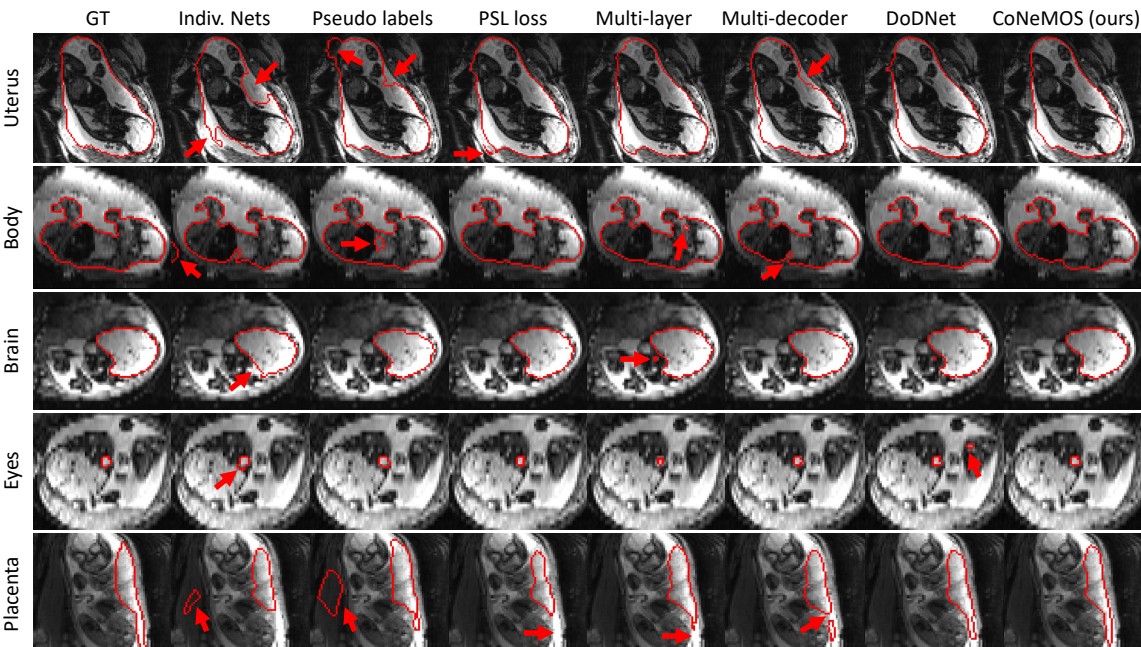

Figure 3: Sample segmentations for all methods and regions. Arrows point at major errors. Regions can be nested (e.g., uterus-body-brain) or disjoint (e.g., placenta-uterus).

Table 2: Comparison of parameters, training time, and inference time.

|  | Indiv. nets | Pseudo lab. | PSL loss | Multi-layer | Multi-decod. | DoDNet | CoNeMOS |
|---|---|---|---|---|---|---|---|
| # Parameters | 7.31M | 1.46M | 1.46M | 1.47M | 3.79M | 1.48M | 1.52M |
| Training time (h) | 120 | 144 | 24 | 24 | 72 | 24 | 24 |
| Inference time (s) | 1.727 | 0.233 | 0.233 | 0.244 | 0.436 | 0.380 | 0.503 |

## 4.2. Results

**Comparison to the state-of-the-art.** Here, we compare CoNeMOS against all baselines (Figure 2, Appendix B). First, individual networks are largely outperformed by all other approaches, which highlights the benefits of leveraging cross-label synergies with *unified* models. In contrast, CoNeMOS presents a good compromise between memory, training, and inference times (Table 2), and yields consistently better results than all baselines, with average Dice improvements ranging from 1.8 (DoDNet) to 3.3 (Pseudo labels). The increase in Dice is best seen for the eyes and placenta, where our method beats all competitors by at least 2.1 and 1.6 Dice, respectively. While CoNeMOS arguably yields better predictions for the uterus and body (Figure 3), this only translates to modest gains in Dice metric, for which misclassified voxels are less penalised than for small or thin organs (eyes, placenta). Yet, clear improvements can be seen in SD95 (95$^{\text{th}}$ percentile of the surface distance) for the uterus and body, where CoNeMOS yields the best results by 0.7 voxel. More generally, the superior robustness of our method is shown by its consistently better SD95 (e.g., 1.1 voxels on average vs. DoDNet) and much fewer outliers (e.g., SD95 for the eyes, Dice for the brain).

**Simulated smaller training datasets.** In this experiment, we evaluate the performance of all methods as a function of the training set size. Studying the robustness of partially

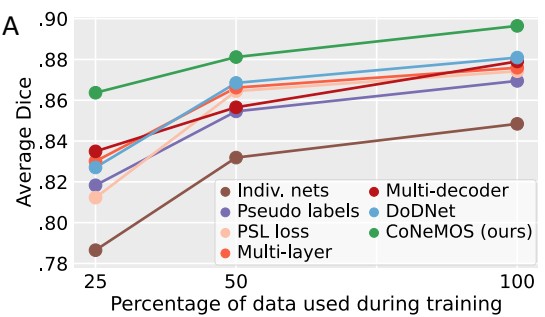
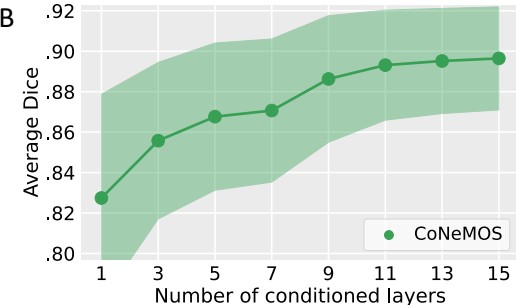

Figure 4: Average Dice obtained by retraining: (A) all methods on simulated smaller datasets, (B) CoNeMOS with increasing numbers of conditioned convolutions ($\pm$ std).

supervised methods to smaller training sets is particularly relevant in the context of reducing the annotation labour. Here, we retrain all methods on 100, 50 and 25% of the training dataset. Importantly, CoNeMOS maintains a high accuracy, as it only loses 3.6 Dice and yields average scores above 86 in all cases (Figure 4A). This is better than any baseline; for instance, DoDNet loses 5.1 Dice across the same range. Overall, CoNeMOS demonstrates robustness to small datasets, which is a distinct advantage compared to its competitors.

**Number of conditioned layers.** Finally, we study the accuracy of CoNeMOS as a function of the number of conditioned convolution layers. We start from the last layer of the CNN (N=1), and progressively condition the entire network (i.e., N=15). Figure 4B shows the drastic increase in accuracy with the number of conditioned layers (+7.2 Dice). This suggests that conditioning all layers is highly beneficial as it gives the network the flexibility to learn where features should be shared or kept label-specific. Learning such representations might require complex interactions at different stages of the network, which cannot be captured with fixed architectures such as shared encoders or decoders.

## 5. Conclusion

We have presented CoNeMOS, a novel conditional framework for partially supervised learning with region-based segmentations. This was achieved by dynamically modulating a segmentation network with label-conditioned FiLM layers in order to encourage cross-label synergies while benefiting from label-specific information. As opposed to previous methods, we condition the *entire* network, such that it can flexibly learn at which stages label-specific information is the most relevant. As a result, CoNeMOS learns robust region representations, which enables it to outperform state-of-the-art methods by a substantial margin. Importantly, our method maintains a good performance level when data becomes less available, which is particularly desirable for partial supervision with the aim of reducing the annotation effort. Future work will focus on integrating the proposed strategy within semi-supervised frameworks to further improve performances by leveraging unlabelled datasets. By relaxing the requirements in fully supervised data, our work promises to facilitate the training and deployment of neural networks for research studies and clinical practice.

**Acknowledgements.** Research supported by NIH NIBIB (P41EB015902, 5R01EB0327-08), NIH NICHD (R01HD100009, R00HD101553), and NSF (2321684).

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

## Appendix A. PSL loss

Let us consider a training image $X_n$ and its corresponding ground truth segmentation $Y_n$, which is only annotated for a subset $\mathfrak{L}_n$ of all target labels $\{1, ..., K\}$. As in Section 3.1, we consider that $Y_n$ is a multi-channel label map $Y_n = \{Y_{nk}\}_{k=1}^K$, where $Y_{nk}$ is a binary segmentation if $k \in \mathfrak{L}_n$, and is empty otherwise. If $\hat{Y}_n = \{\hat{Y}_{nk}\}_{k=1}^K$ is a predicted segmentation, the PSL-aware loss $\mathcal{L}_{\mathrm{PSL}}$ is implemented as:

$$\mathcal{L}_{\mathrm{PSL}} = \frac{1}{|\mathfrak{L}_n|} \sum_{k \in \mathfrak{L}_n} \frac{2 \times \sum_i Y_{nk,i} \times \hat{Y}_{nk,i}}{\sum_i Y_{nk,i}^2 + \sum_i \hat{Y}_{nk,i}^2}, \tag{A1}$$

where $i$ indexes voxels, and $|\mathfrak{L}_n|$ denotes the cardinality of $\mathfrak{L}_n$. In other words, $\mathcal{L}_{\mathrm{PSL}}$ is the soft Dice loss computed only on the available annotated regions for image $X_n$.

## Appendix B. Detailed comparison of all methods

Dice scores (in %) and SD95 (95th percentile of surface distance) for all methods evaluated on the testing set. Standard deviation are in parentheses. The best score is in bold for each region. Stars $*$ denote statistical significance at a 5% level (Bonferroni-corrected Wilcoxon signed rank test). We emphasise that our method (CoNeMOS) yields the best scores for nearly all regions and metrics.

| | | Indiv. nets | Pseudo lab. | PSL loss | Multi-layer | Multi-decod. | DoDNet | CoNeMOS |
|---|---|---|---|---|---|---|---|---|
| Uterus (N=5) | Dice | 92.1 (3.3) | 93.1 (0.9) | 94.1 (0.9) | 93.1 (1.1) | 94.2 (1.4) | 94.6 (0.8) | **95.3 (0.9)** |
| | SD95 | 4.5 (3.5) | 3.9 (1.9) | 3.3 (1.2) | 3.5 (1.3) | 2.9 (1.2) | 2.4 (0.6) | **1.7 (0.6)** |
| Body (N=5) | Dice | 89.3 (6.0) | 92.4 (1.5) | **93.8 (1.2)** | 92.2 (1.5) | 92.2 (1.3) | 92.7 (2.8) | 93.6 (1.5) |
| | SD95 | 4.1 (2.4) | 2.8 (1.0) | 2.3 (0.6) | 2.6 (0.9) | 2.7 (1.1) | 2.1 (0.5) | **1.4 (0.4)** * |
| Brain (N=32) | Dice | 92.2 (4.4) | 93.4 (1.7) | 94.4 (1.3) | 94.6 (1.4) | 94.1 (1.8) | 94.6 (1.5) | **95.6 (1.1)** * |
| | SD95 | 1.8 (1.3) | 1.3 (0.4) | **1.2 (0.3)** | 1.3 (0.4) | 1.3 (0.4) | 1.4 (0.5) | **1.2 (0.3)** |
| Eyes (N=29) | Dice | 75.3 (12.5) | 80.8 (8.0) | 82.8 (8.5) | 82.6 (7.1) | 82.8 (8.5) | 80.1 (9.3) | **84.9 (6.1)** * |
| | SD95 | 2.3 (1.7) | 2.0 (1.6) | 1.4 (0.8) | 1.8 (1.3) | 1.8 (1.5) | 2.5 (1.8) | **1.3 (0.4)** * |
| Placenta (N=44) | Dice | 76.1 (10.6) | 77.9 (8.1) | 80.8 (7.5) | 81.0 (7.6) | 81.7 (6.1) | 83.1 (7.5) | **84.7 (3.8)** * |
| | SD95 | 10.3 (6.3) | 7.3 (5.7) | 6.6 (5.1) | 5.3 (4.7) | 6.6 (5.1) | 6.2 (4.3) | **3.7 (2.2)** * |

