# OpenReview forum: "Network conditioning for synergistic learning on partial annotations"
_MIDL.io/2024/Conference — MIDL 2024 Oral_

### Official Review · Reviewer_GmwJ · 2024-02-28

**Confidence:** 4
**Preliminary Rating:** 5
**Recommendation:** Oral
**Final Rating:** 5

**Summary:**

The authors use a novel strategy to deal with partially annotated medical data. The use of so-called FiLM layers (feature-wise linear modulation layer) in combination with classical convolutional layer, tuned/trained with an auxiliary MLP is the main contribution of this study. The authors show that the importance of features for specific organs are learnt and not explicitly represented by segmentation heads. In addition, there is strong evidence for constant performance (on par or outperforming) on different organs and tissues in their benchmark dataset.

**Strengths:**

In general, the paper has multiple strengths. I like the general methodological concept/paper idea. The problem is mathematically nicely approached, as far as I can tell the math seems to be correct. The authors use strong baselines and competitive concepts (e.g. multi-layer and multi-decoder strategies), and could show in several experiments and analyses how well their method (CoNeMOS) performs. I like also the analysis of needed data and the amount of pre-conditioned layers, showing the importance of the effect on the full 3D UNet architecture. I also checked the Github code briefly, and it seems to be functional and commented.

**Weaknesses:**

The only thing I don't really understand and to the best of my knowledge it is not adequately discussed, is the effect of not integrating the non-annotated classes into the loss function. In my research, we just ignore pixels that have no label (i.e. explicitly organ A, organ B, or background) in the loss computation. This also leads to very accurate results on unseen data. I would be happy if the authors can address this in the discussion and/or introduction section.

**Detailed Comments:**

In Figure 3, I find it hard to compare GT to the "ours", maybe the authors could try to repeat GT on the right side of the Figure to have a better comparison and see the differences better as GT <-> ours and ours <-> competitors is the better comparison.

In Figure 4B, please provide error bars if possible (std? sem?), in Fig 4A it will be too convoluted.

**Justification Of Final Rating:**

My points have been addressed by the authors and I still believe that this study needs to be presented at MIDL. And now I am looking for another tens of characters to make my point in the justification of this review.

**Justification Of The Preliminary Rating:**

I believe the study is of great interest to the community, the paper is well written and proper analyses have been performed. I like the idea and the comparison to strong baselines. I would be very happy to see this at the conference.

**Questions To Address In The Rebuttal:**

Don't need to change my mind. I'm going for an accept.

**Special Issue:**

Yes

---

> ### Author Response · Authors · 2024-03-15
> **Response to reviewer GmwJ**
>
> **1. Why not simply ignore pixels that have no label in the loss computation?** \
> We kindly point out to the reviewer that the suggested strategy corresponds exactly to the PSL-Loss baseline, which is outperformed by our method by 2.5 Dice points on average (Fig. 2), and is less robust to smaller datasets (see Fig. 4A). We have clarified this point by adding a precise description of the PSL-Loss baseline in Appendix A.
>
> **2. Adding error bars in Fig. 4B.** \
> We thank the reviewer for the suggestion and have added standard deviations in Fig. 4B. Overall, the standard deviations decrease from 5.13 (only one convolution is conditioned) to 2.55 (all convolutions are conditioned).
>
> **3. Repeating GT column in Fig. 3.** \
> Following the reviewer’s suggestion, we tried repeating the GT column, but unfortunately, it made the figure too cramped to interpret easily. We would be happy to consider further suggestions in the final review.

---

> > ### Comment · Reviewer_GmwJ · 2024-03-16
> >
> > Ah okay, thank you! Yes, I then didn't get the baseline loss, but here you go. Thank you for your rebuttal, looking Forward for the presentation at MIDL.

---

### Official Review · Reviewer_mtcN · 2024-02-29

**Confidence:** 4
**Preliminary Rating:** 5
**Recommendation:** Oral
**Final Rating:** 5

**Summary:**

The paper proposes a new method which performs segmentation from partially labelled super-region annotations. This is achieved by incorporating FiLM layers in the network.

**Strengths:**

* Novel application of FiLM layers for a clinically relevant task.
* Code is already publicly available.
* Clear description of related work for PSL and what is missing from them.
* Contributions are explicitly written
* Really good ablation study with different percentages of labelled data and different numbers of conditioning layers.

**Weaknesses:**

* The proposed method utilizes the FiLM layer. AdaIN [1] also works in a similar direction but in addition to the FiLM layer, it also incorporates normalizing parameters. AdaIN has been used in various computer vision [2,3] and medical imagining tasks [4,5]. It might be a good idea to compare the proposed method where the FiLM layer is replaced with AdaIN and see if leads to better performance.
* Authors mention that "Alternatively, partial supervision can be addressed by volumetric regularisation (Zhou et al., 2019). While this method is compatible with region-based segmentation (Ulrich et al., 2023), it might not be suited to highly variable anatomies, such as fetuses.". Can authors give a bit more rationale behind this? Why Zhou et al., 2019 might not be suited for fetuses?
* It might be beneficial to perform statistical tests across different methods to show that the proposed method indeed gives better results.

[1] Huang, X. and Belongie, S., 2017. Arbitrary style transfer in real-time with adaptive instance normalization. In Proceedings of the IEEE international conference on computer vision (pp. 1501-1510).

[2] Zhang, Y., Ling, H., Gao, J., Yin, K., Lafleche, J.F., Barriuso, A., Torralba, A. and Fidler, S., 2021. Datasetgan: Efficient labeled data factory with minimal human effort. In Proceedings of the IEEE/CVF Conference on Computer Vision and Pattern Recognition (pp. 10145-10155).

[3] Nam, H. and Kim, H.E., 2018. Batch-instance normalization for adaptively style-invariant neural networks. Advances in Neural Information Processing Systems, 31.

[4] Jacenków, G., O’Neil, A.Q., Mohr, B. and Tsaftaris, S.A., 2020. INSIDE: steering spatial attention with non-imaging information in CNNs. In Medical Image Computing and Computer Assisted Intervention–MICCAI 2020: 23rd International Conference, Lima, Peru, October 4–8, 2020, Proceedings, Part IV 23 (pp. 385-395). Springer International Publishing.

[5] Nichyporuk, B., Cardinell, J., Szeto, J., Mehta, R., Falet, J.P.R., Arnold, D.L., Tsaftaris, S.A. and Arbel, T., 2022. Rethinking Generalization: The Impact of Annotation Style on Medical Image Segmentation. arXiv preprint arXiv:2210.17398.

**Detailed Comments:**

No Minor comments. Paper is well written with enough details and good formating.

**Justification Of Final Rating:**

I am happy with the response provided by the authors. I would recommend acceptance of the paper. Looking forward to the presentation at the conference, and hopefully a journal extension with more detailed analysis.

**Justification Of The Preliminary Rating:**

Overall, the paper is well written, with clear motivation, experimental results, and ablation study. I do not have many major comments and would recommend direct acceptance of the paper. I think, the MIDL community would benefit from such papers as it will lead to further conversations regarding partially labelled segmentation scenarios.

**Questions To Address In The Rebuttal:**

Address some of the comments mentioned in the weakness section.

**Special Issue:**

Yes

---

> ### Author Response · Authors · 2024-03-15
> **Response to reviewer mtcN**
>
> **1. Consider replacing FiLM layers with AdaIN layers.** \
> We thank the reviewer for pointing this out. In our early experiments, we tested replacing FiLM by AdaIN layers, but this resulted in slightly lower scores (-0.6 Dice points on average). This is consistent with reference [5] provided by the reviewer (which actually uses FiLM and not AdaIN). We have included AdaIN in the Related work section.
>
> **2. Explanation on why volumetric regularisation (Zhou et al., 2019) is not suitable for fetuses.** \
> We agree that this needed further explanation. The volumetric regularisation used in Zhou et al. aims to penalise predicted segmentations with volumes that differ substantially from the population statistics. This implicitly assumes that a target region’s volume varies very little across the studied population. This assumption does not hold in fetuses due to fetal growth and pregnancy maturation, where region size varies substantially across the studied gestational age span (19 to 35 weeks). We clarified this point in the paper (p.3).
>
> **3. Adding statistical tests.** \
> We thank the reviewer for the suggestion and have performed two-sided Bonferroni-corrected non-parametric Wilcoxon signed-rank tests. We added the results to , Fig. 2 and to the corresponding new table in Appendix B. Importantly, these tests show that CoNeMOS significantly outperforms the other baselines at a 5% level for the Body (SD95), Brain (Dice), Eyes (Dice, SD95), and Placenta (Dice, SD95).

---

### Official Review · Reviewer_QNL9 · 2024-02-29

**Confidence:** 4
**Preliminary Rating:** 4
**Final Rating:** 5

**Summary:**

This paper propose a method baed on network conditioning, to deal with semantic segmentation when only partial label are available (i.e.: many objects can appear in the images, but only a few are labeled on a single image. This makes it difficult to supervise the background and false positives, as other object could be in the "background" of the annotations.)

The method is elegant and scalable, and the results seem very good, albeit evaluated on a single dataset.

**Strengths:**

- The paper is very well written
- The related works seem thorough, and the discussion around it is helpful, motivating the proposed method
- Comparison with many methods
- both overlap and distance based metrics are used
- sensitivity study wrt amount of available data, and number of conditioned layers.

**Weaknesses:**

Aside from the evaluation performed on a single dataset (albeit with multiple organs), I see no significant weakness at this time ‎ ‎ ‎ ‎ ‎ ‎ ‎ ‎ ‎ ‎ ‎ ‎ ‎ ‎ ‎ ‎ ‎ ‎ ‎ ‎ ‎ ‎ ‎ ‎ ‎ ‎ ‎ ‎ ‎ ‎ ‎ ‎ ‎ ‎ ‎ ‎ ‎ ‎ ‎ ‎ ‎ ‎ ‎ ‎ ‎ ‎ ‎ ‎ ‎ ‎ ‎ ‎ ‎ ‎ ‎ ‎ ‎ ‎ ‎ ‎ ‎ ‎ ‎ ‎ ‎ ‎ ‎ ‎ ‎ ‎ ‎ ‎ ‎ ‎ ‎ ‎ ‎ ‎ ‎ ‎ ‎ ‎ ‎ ‎ ‎ ‎ ‎ ‎ ‎ ‎ ‎ ‎

**Detailed Comments:**

The authors should add in the appendix the full table with their results.

Writting the PSL aware loss(es?) in the appendix could be helpful, especially if the reviewer isn't able to download other papers when reading the paper.

**Justification Of Final Rating:**

I thank the authors for their detailed response, I am very happy with it.

I increased my score to strong accept, and it seems that I cannot click now the "oral/special issue/consider for best paper award" buttons, but the intention is there.

Great work, looking for the talk in Paris! (Is my previous sentence an oxymoron? I don't know.)

**Justification Of The Preliminary Rating:**

I am always conservative in my initial rating, but I would rate it as 4.5 rather than 4. Once more information is available around the complexity (see questions for rebuttal) it could easily be upgraded. Oral/special issue recommendation will come after rebuttal.

**Questions To Address In The Rebuttal:**

I would like the authors to discuss the speed and complexity (flops, memory, convergence speed) of all methods. Notably, unless I missed it, there is no discussion of the added cost of the conditioning-MLP. At the same time, I assume it saves memory compared to other methods (notably indiv. networks and multiple decoders) so this should be discussed.

SD95 is reported in voxels, but is it computed in 3D or in 2D? Do all axises have the same spatial resolution? If not I would have expected that metric to be in an actual distance unit (such as micro-furlong).

Is there a specific reason why the eyes and placenta results are much lower (and variable) than the other organs (Fig 2, DSC). I can easily image the placenta having a poor contrast and very variable shape, making it difficult, but I am more surprised by the eyes (that have a consistent shape). Or am I misreading the results?

---

> ### Author Response · Authors · 2024-03-15
> **Response to reviewer QNL9**
>
> **1. Speed and complexity comparison.** \
> We thank the reviewer for the suggestion, and have added a new table summarising the memory requirements and training/inference speed (see below and p. 7 in the paper). It shows that CoNeMOS outperforms all baselines without needing extra training time, using roughly the same number of parameters (4% increase vs. a UNet, and 50% decrease vs. Multi-decoders), all while keeping a reasonable inference time (sub-second).
>
> |  | Indiv Nets. | Pseudo lab. | PSL loss | Multi-layer | Multi-decod. | DoDNet | CoNeMOS |
> |-|-|-|-|-|-|-|-|
> | # Parameters | 7.31M | 1.46M | 1.46M | 1.47M | 3.79M | 1.48M | 1.52M |
> | Training time (h) | 24x5 | 24x5+24 | 24 | 24 | 72 | 24 | 24 |
> | Inference time (s) | 1.727 | 0.233 | 0.233 | 0.244 | 0.436 | 0.380 | 0.503 |
>
> **2. Is SD95 computed in 2D or 3D? If all axes do not have the same resolution, this should be measured with a distance unit, rather than voxels.** \
> This metric was computed in 3D. In our datasets, all axes have the same spatial resolution (3mm isotropic), as mentioned in Section 4.1/Dataset.
>
> **3. Why are Dice scores lower for the eyes and placenta?** \
> We thank the reviewer for pointing this out. The placenta is hard to segment because it has a very variable shape and can present poor tissue contrasts. Moreover, the placenta is a thin structure for which even small segmentation errors result in a significant drop in the Dice score. The lower Dice scores for the eyes are similarly explained by their much smaller size compared to the other regions, which means that mis-segmented voxels impact Dice scores much more severely. This is partly why we also report SD95, which is much more robust to region size. We have clarified this point in the revised paper (p.7).
>
> **4.Adding a description of the PSL-aware loss in the appendix.** \
> The PSL-aware loss used in our experiments simply amounts to a soft Dice loss computed only on the available labels for each training scan. We have added the precise description of this loss to Appendix A.
>
> **5. Adding a full result table to the appendix.** \
> Following the reviewer‘s suggestion, we have added a table that reports the full set of results to Appendix B.

---

### Author Response · Authors · 2024-03-15
**Overall response**

We thank the reviewers for their time and insightful comments, which have helped us strengthen our submission. We are happy that they found our method novel and clinically relevant, well-explained, and supported by convincing results (comparison against strong baselines and insightful ablations).

In response to reviewers’ comments, we have added:
- a comparison of memory and speed requirements of all tested methods [reviewer QNL9],
- statistical tests in experiment 1, which shows that our method significantly outperforms baselines in the large majority of metrics/segmented regions [reviewer mtcN]
- error bars for the results of the ablation study on the number of conditioned layers [reviewer GmwJ].

Beyond these changes, we have addressed all comments raised by the reviewers (see individual responses below), and edited the manuscript accordingly (changes in red in the revised submission). Again, we deeply appreciate the feedback and would be happy to address any remaining questions during the next phase.

---

### Meta-Review · Area_Chair_iCBy · 2024-04-03

**Recommendation:** Accept (Oral)
**Confidence:** 5

**Metareview:**

This paper proposes a method for handling partially labeled data in multi-organ segmentation by using FiLM layers to condition the network on the target label.

Strengths:
+ Very well written paper
+ Clear motivation and reasoning for approach
+ Strong experiments with comprehensive ablation/sensitivity studies and comparisons to related methods
+ Code is already shared

Weaknesses:
- Evaluation performed on a single dataset
- Did not present comparison with related AdaIN method

All reviewers are in agreement that this is a very strong paper and recommend oral/special issue. I concur with the reviewers and recommend accept.

---

### Decision · Program_Chairs · 2024-04-06

Accept (Oral)